# Positive attitudes towards COVID-19 vaccines: A cross-country analysis

**Talita Greyling[1]⊗, Stephanié Rossouw[2]⊗***

**1** School of Economics, College of Business and Economics, University of Johannesburg, Gauteng, South Africa, **2** School of Social Science & Public Policy, Faculty of Culture and Society, Auckland University of Technology, Auckland, New Zealand

⊗ These authors contributed equally to this work.
* stephanie.rossouw@aut.ac.nz

**Data Availability Statement:** All relevant data has now been uploaded as part of the Supporting information files.

**Funding:** The author(s) received no specific funding for this work.

## Abstract

COVID-19 severely impacted world health and, as a consequence of the measures implemented to stop the spread of the virus, also irreversibly damaged the world economy. Research shows that receiving the COVID-19 vaccine is the most successful measure to combat the virus and could also address its indirect consequences. However, vaccine hesitancy is growing worldwide and the WHO names this hesitancy as one of the top ten threats to global health. This study investigates the trend in positive attitudes towards vaccines across ten countries since a positive attitude is important. Furthermore, we investigate those variables related to having a positive attitude, as these factors could potentially increase the uptake of vaccines. We derive our text corpus from vaccine-related tweets, harvested in real-time from Twitter. Using Natural Language Processing (NLP), we derive the sentiment and emotions contained in the tweets to construct daily time-series data. We analyse a panel dataset spanning both the Northern and Southern hemispheres from 1 February 2021 to 31 July 2021. To determine the relationship between several variables and the positive sentiment (attitude) towards vaccines, we run various models, including POLS, Panel Fixed Effects and Instrumental Variables estimations. Our results show that more information about vaccines' safety and the expected side effects are needed to increase positive attitudes towards vaccines. Additionally, government procurement and the vaccine rollout should improve. Accessibility to the vaccine should be a priority, and a collective effort should be made to increase positive messaging about the vaccine, especially on social media. The results of this study contribute to the understanding of the emotional challenges associated with vaccine uptake and inform policymakers, health workers, and stakeholders who communicate to the public during infectious disease outbreaks. Additionally, the global fight against COVID-19 might be lost if the attitude towards vaccines is not improved.

## 1. Introduction

In an attempt to curb the spread of COVID-19, minimise the loss of life and take the pressure off the national health systems, governments worldwide started their vaccine rollout

**Competing interests:** The authors have declared that no competing interests exist.

campaigns late in December 2020. However, this rapid rollout of the COVID-19 vaccine has created different emotional responses across the globe. This is problematic since receiving the COVID-19 vaccine is the best possible solution to open up economies and prevent further loss of life. Compounding the problem is the mistrust in governments' abilities to procure and administer the rollout of vaccines and the spread of fake news by anti-vaxxers (for example, see Sharma [1] and Bonnevie et al. [2]). Spreading fear and anxiety is a significant problem because we know from the existing literature that vaccine efficacy depends not only on the vaccine, but also on the characteristics of the vaccinated (Madison et al. [3], Glaser et al. [4]). Unfortunately, the COVID-19 pandemic has already led to increased depression, loneliness, and stress levels, increasing the efficacy problem (Madison et al. [3]). Adding to this, vaccine hesitancy is growing worldwide and is seen as one of the top ten threats to global health. This makes it easy to see that governments today face a significant challenge.

To this end, our primary aim is to conduct a cross-country panel analysis to investigate the trend in positive attitudes towards COVID-19 vaccines over time. This will enable us to determine whether people are becoming more or less positive towards accepting the vaccine and likely reasons for these trends. A secondary aim lies with determining those variables which are significantly related to a positive vaccine attitude and can inform policymakers.

Previous studies (Lyu et al. [5], Xue et al. [6], Chopra et al. [7]) analysed the emotions in vaccine-related tweets. However, their primary aim was to better understand the public perceptions, concerns, and emotions related to *COVID-19 vaccine topics and discussions* on social media. They determined the sentiments related to topics and discussions and investigated the strength of discussions and sentiments over time. The main limitations of these studies include that they: i) only analysed English tweets, with no attention being paid to specific geographical areas or comparing the sentiment across different countries, ii) did not use sentiment analysis in further analyses, and iii) did not investigate the variables related to positive vaccine attitudes. We overcome these limitations by constructing a daily time-series called the Vaccine Positive Attitude Index (VPAI), a real-time measure of people's positive attitudes toward the COVID-19 vaccine across ten countries for the period 1 February 2021–31 July 2021. The countries span both the Northern and Southern hemispheres and include Australia, Belgium, Germany, Great Britain, France, Italy, the Netherlands, New Zealand, South Africa and Spain.

We derive the VPAI using Big Data by extracting a live stream of tweets for specific geographical areas which contains a list of vaccine-related keywords. After the data is cleaned, we use Natural Language Processing (NLP) to derive the sentiment and emotions of the tweets. After calculating the mean levels of positive sentiment per day, we investigate the trend over time in the VPAI and compare it across our panel of ten countries under investigation. Additionally, we determine which variables are related to the VPAI and, therefore, when addressed, could create a more positive attitude and increase the uptake of COVID-19 vaccines. To limit the effect of confounding factors, we introduce various estimation techniques to address possible endogeneity. We use Pooled Ordinary Least Squares as a base model and extend the analyses to include Panel Fixed Effects and Instrumental Variables regressions.

Our results indicate that the VPAI trends downward over time for the whole sample. We find the same results considering the Northern and Southern hemisphere subsamples. Considering the trends per country, we find in all countries a downward trend except Belgium and the Netherlands, which shows a slightly positive trend. Therefore, interventions are needed to change the attitude toward vaccines and increase the uptake. Our results show those variables that can improve the positive attitude towards the COVID-19 vaccines are information-related to the safety and expected side effects of the vaccines, improving trust in vaccines, reviewing regulations implemented to limit the spread of the vaccines as it seems that people weigh-up the benefits of being vaccinated against lockdown regulations. Additionally, increasing trust in

governments to procure and effectively roll out vaccinations should be a priority. Furthermore, social media platforms, such as Twitter, should launch targeted campaigns focusing on educating people about the safety of vaccines, providing progress on the rollout and encouraging all ages to get vaccinated. We are confident that if the factors found significant in the econometric models (confidence levels of 95 per cent or more) are addressed, the positive attitudes towards vaccines will improve. Policy interventions in line with these recommendations will contribute to the universal plan to restore global health and the world economy.

The rest of the paper is structured as follows. The next section contains a brief background of the countries used in our analyses and studies on COVID-19 vaccine hesitancy. Section 3 describes the data and the selected variables, and outlines the methodology used. The results and discussion follow in section 4, while the paper concludes in section 5.

## 2. Background and literature review

### 2.1 Country background

This study focuses on three Southern hemisphere countries; South Africa, New Zealand and Australia and seven Northern hemisphere countries; Belgium, Germany, Great Britain, France, Italy, the Netherlands, and Spain. Primarily the choice of countries is determined by data availability. However, the dataset can be extended to include more countries in future studies. The current selection of countries from both hemispheres provides unique insights into people's attitudes to the COVID-19 vaccine. Table 1 summarises key facts for each country used in the current study.

From Fig 1, we can see that the country performing the worst in terms of the total number of people fully vaccinated is New Zealand (approximately 750,000 people). However, if we consider the vaccinated as a percentage of the total population, South Africa performs the worst

**Table 1. Key summary facts of countries in this study.**

| Country | Total population | Average happiness levels** (2020) | Oxford Stringency Index (Average for the period) | First confirmed COVID-19 case (2020) | Date of first lockdown (2020) | Total confirmed COVID-19 cases (28 August 2021) | Total confirmed COVID-19 deaths (28 August 2021) | Date of vaccine rollout | Percentage of the population fully vaccinated (31 July 2021) |
|---|---|---|---|---|---|---|---|---|---|
| Australia | 25.5 million | 7.09 | 58.64 | 25 January | 17 March* | 51,256 | 999 | 22 February 2021 | 15% |
| Belgium | 11.6 million | 6.98 | 58.30 | 4 February | 13 March | 1.18 million | 25,360 | 28 December 2020 | 60% |
| France | 66.99 million | 6.66 | 63.15 | 24 January | 17 March | 6.81 million | 114,506 | 27 December 2020 | 48% |
| Germany | 83.02 million | 7.08 | 72.71 | 27 January | 22 March | 3.93 million | 92,136 | 27 December 2020 | 52% |
| Great Britain | 66.65 million | 7.17 | 66.70 | 31 January | 23 March | 6.73 million | 132,699 | 8 December 2020 | 56% |
| Italy | 60.36 million | 6.39 | 74.90 | 30 January | 9 March | 4.52 million | 129,002 | 27 December 2020 | 52% |
| Netherlands | 17.28 million | 7.73 | 64.88 | 27 February | 15 March¶ | 1.97 million | 18,339 | 6 January 2021 | 54% |
| New Zealand | 5.5 million | 7.14 | 26.80 | 28 February | 26 March | 3,465 | 26 | 19 February 2021 | 15% |
| South Africa | 57.7 million | 6.32 | 51.90 | 6 March | 27 March | 2.76 million | 81,461 | 17 February 2021 | 5% |
| Spain | 46.94 million | 6.40 | 64.33 | 31 January | 14 March | 4.83 million | 84,000 | 27 December 2020 | 58% |

* Australia never officially went into a complete lockdown such as that seen in the other countries. We used the day when the closure of international borders was announced as a proxy for "lockdown."

¶ The Netherlands started a so-called 'intelligent lockdown' on this date.

** The happiness scores cited here reflect the average for the period in 2020 before the first COVID-19 case was announced.

Sources: Hale et al. [8], Greyling et al. [9], Google [10, 11], Roser et al. [12], Mathieu et al. [13].

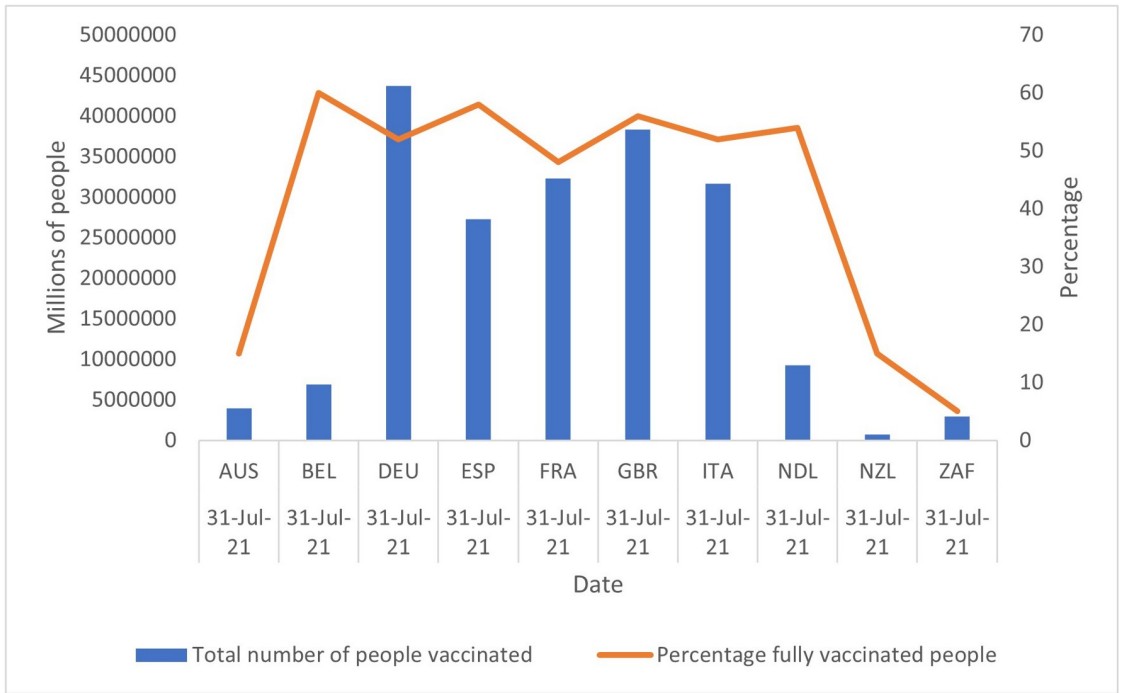

**Fig 1. COVID-19 number of people vaccinated and the percentage of fully vaccinated people per country (31 July 2021).**
Source: Mathieu et al. [13].

with 5 per cent as of 31 July 2021. Of interest is the Northern-Southern hemisphere split. The Northern hemisphere outperforms all three the Southern hemisphere countries (Australia, New Zealand and South Africa). Of the Northern hemisphere countries, France is the worst performer with 48 per cent fully vaccinated, whereas Belgium is the best performer (60 per cent) (Mathieu et al. [13]).

## 2.2 Literature on COVID-19 vaccine hesitancy

There is an exponential growth of studies in the literature on COVID-19 vaccine hesitancy as researchers from all disciplines addresses one of the biggest global health threats.

Research regarding COVID-19 vaccine hesitancy spans across both *online surveys* (see, for example, Akarsu et al. [14], Fisher et al. [15], Freeman et al. [16], Ward et al. [17], Seale et al. [18]) and *in-person surveys* (see, for example, Paul et al. [19], Sallam [20]). Primarily these studies found people's hesitancy and refusal of the COVID-19 vaccine were mostly attributed to i) fear driven by possible side effects of the vaccine, and ii) the unreliability of what is seen as a new vaccine. Paul et al. [19] conducted a study involving surveying 32,361 participants from 7 September to 5 October 2020. The authors found that distrustful attitudes towards vaccination were higher amongst individuals from ethnic minority backgrounds, with lower levels of education, lower annual income, poor knowledge of COVID-19, and poor compliance with government COVID-19 guidelines. Apart from the Paul et al. [19] study, the other aforementioned studies found willingness to take the COVID-19 vaccine was closely related to one's sense of collective responsibility and campaigning for the 'greater good'. Furthermore, these studies highlighted a need for better and more transparent information, the role of anti-vaccination campaigns, and a lack of trust in the government. Interestingly, it was found that low

rates of COVID-19 vaccine acceptance were reported in the Middle East, Russia, Africa and several European countries.

Our current study uses *Big Data* to construct a Vaccine Positive Attitude Index (VPAI); therefore, the rest of the literature review will focus on those that also use Big Data, with special attention to three studies closest to ours in spirit. We note there is a burgeoning of literature using Big Data in the form of Twitter to analyse vaccine-related topics. Therefore, we cannot possibly discuss all of them.

For example, Yousefinaghani et al. [21] used vaccine-related tweets to track frequent hashtags, frequent mentions, main keywords, and main themes with positive and negative sentiments in the tweets. Hussain et al. [22] used Facebook and Twitter to study people's hesitancy, perceptions and sentiment towards the COVID-19 vaccine. Küçükali et al. [23], Nuzhath et al. [24], Bonnevie et al. [2] and Thelwall et al. [25] all identified prominent themes about vaccine hesitancy and refusal on social media during the COVID-19 pandemic. These studies found that the most frequent themes that elicit a negative sentiment are anti-vaccination, poor scientific processes, conspiracy theories, mistrust of scientists and governments, lack of intent to get a COVID-19 vaccine, freedom of choice, and religious beliefs.

Sharma et al. [1] and Bonnevie et al. [26] focused on using Twitter to identify suspicious coordinated accounts in the dataset to find misinformation campaigns that drive the conversation against getting the COVID-19 vaccine. Based on an analysis of the collective behaviours and activities of accounts, they found that they correspond to a 'Great Reset' conspiracy theory and ten additional themes such as research and clinical trials and vaccine ingredients.

Three studies that come the closest to ours in spirit are Lyu et al. [5], Xue et al. [6] and Chopra et al. [7]. Lyu et al. [5] used 1.5 million English vaccine-related tweets collected between March 2020 and January 2021 and categorised the tweets into 16 topics grouped into five overarching themes. Their results showed that under their first theme called "Opinions and Emotions Around Vaccines and Vaccination", the topic out of all 16 topics that were mostly tweeted was opinions about vaccination. In terms of their sentiment analysis (using the Syuzhet lexicon) they found that, apart from fluctuations throughout the period, the sentiment increased regarding the COVID-19 vaccine. Their emotions analysis (using the NRC lexicon) found trust was the most prevalent emotion, followed by anticipation and fear. They found that before Moderna, one of the first to test their COVID-19 vaccine on humans in April 2020, fear was the most prevalent emotion.

Xue et al. [6] analysed 4 million English vaccine-related tweets using a list of 20 hashtags from 7 March to 21 April 2020. Their main aim was to identify popular unigrams (one word) and bigrams (two words), salient topics and themes, and sentiments in the collected tweets. In terms of unigrams, they found "virus", "lockdown", and "quarantine" to be the most popular. Bigrams "COVID-19", "stay home", "corona virus", "social distancing" and "new cases" was the most popular. Furthermore, they identified 13 discussion topics from the tweets and categorised them into five different themes. For example, theme 1, "Public health measures to slow the spread of COVID-19", included topics such as face masks, quarantine, test kits, lockdown, safety, vaccine and US shelter-in-place. Their emotions analysis (using the NRC lexicon) showed that anticipation followed by fear, trust, and anger were prevalent across 12 of the 13 topics.

Chopra et al. [7] collected 1.8 million English vaccine-related tweets from across India, the United States, Great Britain, Brazil, and Australia from June 2020 to April 2021. They aimed to create ten lexical categories, split between two classes, namely emotions (6 categories) and influencing factors (4 categories) and study the temporal evolution of these categories across time. The lexical emotions category includes hesitation, sorrow, faith, contentment, anticipation and rage, while their influencing factors are misinformation, vaccine rollout, inequities,

and health effects. The authors used the word-count approach to measure each category's strength in a given tweet. They calculated the strength of the categories monthly and split their period under investigation in two; Before and After the date when each country's government approved the first COVID-19 vaccine. Their results differed across countries with, for example, India experiencing a decrease in the strength of hesitation experienced after vaccine approval, with mentions of health effects contributing the most in tweets with a positive hesitation score. The United States experienced a significant increase in contentment after their vaccine approval. Rage and discussions on misinformation became significantly higher after vaccine approval in India, whereas the opposite was true for the United States.

Given the above literature review, no other study has done what we propose to do. We will be the first study to use Big Data to determine the sentiment and emotions related to COVID-19 vaccines through a vaccine positive attitude index. Additionally, no other study has followed the trends in attitudes over time and derived emotion and sentiment time-series data across countries to determine the variables that significantly influence a positive attitude towards the COVID-19 vaccine.

## 3. Data and methodology

### 3.1 Data

In the analyses, we use a cross-country panel dataset with high-frequency daily data (see section 3.2). We analyse the time period from 1 February 2021 to 31 July 2021 (181 days) across ten countries.

**3.1.1 Constructing time-series data using sentiment and emotions analysis.** To derive our time-series data which capture sentiment and emotions, we construct variables using Big Data by extracting tweets from Twitter. In our analysis, we extracted two sets of tweets based on keywords, the one related to COVID-19 vaccines and the other related to government. The tweets containing these words amounted to 1,047,000 tweets. We extracted all tweets according to specific geographical areas (country).

The first step in our analysis is to determine the tweets' language (we detected 64 different languages), and all non-English tweets were translated to English. After the translation process, we use NLP to extract the sentiment and the underlying emotions of the tweets. To test the robustness of the coding of the sentiment of the translated tweets, we use lexicons in the original language, if available, and repeat the process. We compare the coded sentiment of the translated and original text and find the results strongly correlated.

We make use of a suite of lexicons. Each of them differs slightly but with the primary aim to determine the sentiment of unstructured text data. The two lexicons mostly used in our analysis are Sentiment140 and NRC (National Research Council of Canada Emotion Lexicon developed by Turney and Mohammad [27]). The other lexicons are used for robustness purposes and are part of the Syuzhet package. The lexicons include Syuzhet, AFINN and Bing. The sentiment is determined by identifying the tweeter's attitude towards an event using variables such as context, tone, etc. It helps one form an entire opinion of the text. Depending on the lexicon used, the text (tweet) is coded. For example, if a tweet is positive, it is coded as 0, if neutral 2 and if negative 4.

We use the NRC lexicon to code the sentiment (as explained above) and analyse the underlying tweets' emotions. It distinguishes between eight basic emotions: anger, fear, anticipation, trust, surprise, sadness, joy and disgust (the so-called Plutchik [28] wheel of emotions). NRC codes words with different values, ranging from 0 (low) to 8 (the highest score in our data), to express the intensity of an emotion or sentiment.

To construct the time-series data, we use the coding of the tweets and derive daily averages. In this manner, we derive a positive sentiment, a negative sentiment and eight emotion time-series. We derive the sentiment time-series using different lexicons as a robustness test and compare these results using correlation analyses. We perform various additional robustness tests, for example, to determine whether the sampling frequency significantly influences the results.

To test the robustness of the *frequency*, we construct the relevant index (time-series) per day (the norm), we repeat the exercise but construct the time-series per hour. We find similar trends in our hourly and daily time-series, indicating that the timescale at which sampling takes place does not significantly influence the observed trend.

To test whether the *volume* of tweets affects the derived time-series data, we extract random samples of differing sizes from the daily text corpus of tweets. The time-series based on these smaller samples (50 per cent and 80 per cent of the daily extracted tweets) are highly correlated to the original time-series.

## 3.2 Selection of variables

**3.2.1 The outcome variable: Vaccine positive attitude index (VPAI).** To construct the VPAI index, we follow the method explained above and extract COVID-19 related tweets using the keywords: *vaccinate*, *vacc*, *vaccine*, *Sputnik V*, *Sputnik*, *Sinopharm*, *Astrazeneca*, *Pfizer (if NEAR) vaccine*, *Pfizer-BioNTech*, *Johnson & Johnson*, and *Moderna*.

To ensure that the extracted vaccine-related tweets discuss attitudes related to receiving the COVID-19 vaccine, we first constructed word clouds per country. For example, Fig 2 illustrates the word cloud generated for Great Britain. After generating word clouds for all countries, we returned to the original tweets and confirmed the context of the words with high frequencies. We determined that these vaccine-related tweets are indeed related to receiving the vaccine and expressed that "it's good to receive a vaccine" and that people are happy after receiving their second vaccination. For example, tweets that generated the word cloud for Great Britain included:

"*Here it is, worth its weight in gold. My consent form for the covid vaccine next week, normality on the horizon hope*"

"*So excited to hear my mum, an NHS nurse, will be receiving the Pfizer Covid-19 vaccine today—a glimmer of hope af*"

"*Grandmother the vaccine, as you can see, absolutely delighted?? (all credit to my younger brother for this absolute*"

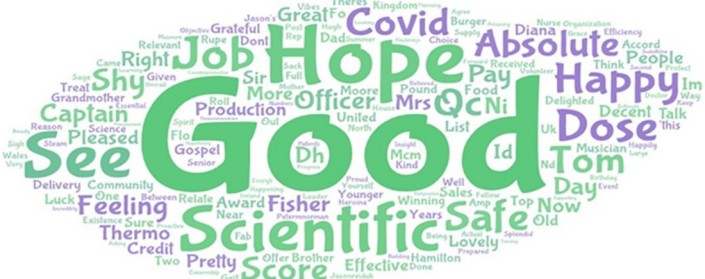

**Fig 2. Word cloud based on positive sentiment for vaccine-related tweets, Great Britain.** Source: Authors own compilation using word cloud software.

Please note the above tweets were taken directly from Twitter and do not represent the views of the authors or their institutions.

From the positive sentiment vaccine-related tweets, we determined that more than 90 per cent were directly related to receiving the COVID-19 vaccine. We realise that the data carries limited noise, but we believe that this noise does not affect our results, especially considering the large number of tweets analysed.

As discussed in section 3.1.1, we use the NRC lexicon to calculate our VPAI by deriving the mean value of the positive sentiment coded tweets per day and standardising these values using the minimum-maximum method. The index is measured on a scale from 0 to 1, with 0 the lowest positive sentiment and 1 the highest. As a robustness test for the VPAI, we derive a similar index using Sentiment140. However, in this instance, we calculate the VPAI by expressing the number of positive tweets per day as a percentage of the sum of the number of positive and negative sentiment tweets (see the supporting information section for the graphs on the trends in positive attitudes using the VPAI based on Sentiment140).

**3.2.2 Selection of covariates.** To select the covariates in the analyses, we used several methods, including relevant literature (see section 2.2), theoretical models, topic modelling and the analysis of negative sentiment and negative emotion tweets.

By analysing the latter, we can determine the major issues that limit the uptake of vaccines and which are likely related to decreased positive attitudes. The reader should note that the negative sentiment associated with vaccine tweets is not the inverse of the positive sentiment. Our analyses find that negatively coded tweets primarily relate to anger, fear or sadness due to the procurement and the efficiency of the vaccine rollout and a lack of information about the side effects. Additionally, our topic modelling revealed that people are dissatisfied with vaccine passports and QR scanning. Concerns were expressed about the Delta variant and misinformation (COVID-19 vaccines and the virus is fake), and various conspiracy theories also came to light, especially for New Zealand. People also express their discontent with social distancing and wearing masks.

In terms of theoretical framework, we use a measure that captures relevant predictors of vaccination behaviour, called the 5C scale. The 5C scale measures the "psychological antecedents of vaccination" as designed by Betsch et al. [29] and is grounded in established theoretical models of vaccine hesitancy and acceptance (Thomson et al. [30], MacDonald [31], Larson et al. [32]) and relates these predictors to psychological models to explain health behaviour (Betsch et al. [33]). We note from the 5C that confidence, constraints, collective responsibility, complacency, and calculation are important when investigating vaccination behaviour.

For the analysis of the negative sentiment and negative emotion vaccine-related tweets, we follow the same process as described in section 3.2.1. For an example of the issues that cause negative sentiment and emotions in people, see the word cloud in Fig 3 generated from tweets extracted for South Africa.

Sample tweets that generated the word cloud for South Africa's negative sentiment include, for example:

"*With an incompetent government, a Minister of Health without a medical degree, NDZs dictatorial tendencies & a rural population still totally unaware of what a pandemic is added to a vaccine shortage, we are doomed*"

*We are bored about 1) corruption 2) poor vaccine strategy 3) terrible national government 4) incompetent cabinet 5) stealing during a pandemic!*"

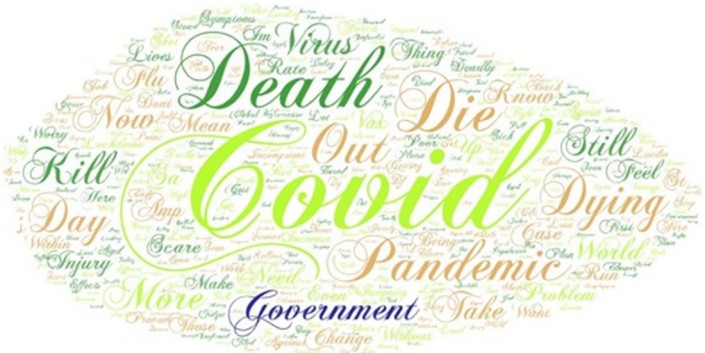

**Fig 3. Word cloud based on negative sentiment for vaccine-related tweets, South Africa.** Source: Authors own compilation using word cloud software.

> *This vaccine rollout has been disastrous from the government, has cost lives, now another lockdown killing our already hurt economy. Massive change is needed in the running of this country*

> *The rate with which people are dying every day should get SAHPRA concerned and energised to approve more vaccine even on trial basis, this apparent incompetence is really killing and destroying families.*

Please note that the above tweets were taken directly from Twitter and do not represent the views of the authors or their institutions.

After conducting an in-depth analysis of the negative sentiment vaccine-related tweets for all ten countries, we discovered that the negative sentiment was mainly related to anger towards governments' incompetence in procurement, the lack of procuring a sufficient number of (or wrong) vaccines and the execution of the vaccine rollout, fear regarding side effects, fear of people dying because they cannot get access to vaccines and people refusing to be vaccinated. Interestingly, the words prominent in the word cloud, such as 'death', 'die', and 'killing', are related to not receiving the vaccine rather than fearing the side effects.

In analysing the negative sentiment tweets, we found that tweets expressing dissatisfaction with governments are false negatives. This means that, in reality, people have positive attitudes towards vaccines. However, they are negative about government incompetence related to issues such as the rollout process, the procurement and accessibility of vaccines etc., hence the negative sentiment of the tweets. To test this hypothesis, we also create a VPAI in which we add the tweets coded as false negatives to the tweets coded as positive. We name the index VPAI2. See the trends in the supporting information section. S4 to S6 Figs indicate a predominant upward trend. This suggests that if policymakers address the grievances of people related to the abovementioned government incompetence, they can turn around the downward trend in the VPAI.

Therefore, the selected covariates included in the regression analyses are:

1. Trust in the COVID-19 vaccine: as a proxy for how people perceive the vaccine's safety. To construct this variable, we follow the method as explained in section 3.3.1. We use the *NRC lexicon* to return the emotion score for each COVID-19 vaccine-related tweet for 'trust'. We construct a daily time-series by averaging the measured value of 'trust' per tweet per day (Greyling et al. [9], Betsch et al. [29]). We lag the variable to address possible endogeneity that might spread from confounding factors.

2. Anger towards the government: is included in our interaction variable (see point 8). However, to construct the 'anger towards the government' variable, we first extract all tweets that include the following keywords: *government*, *parliament*, *ministry*, *minister*, *senator*, *MPs*, *legislator*, *political*, *politics*, *prime minister*. We use the same method to construct the time-series as for the 'trust in the COVID-19 vaccine' variable (Greyling et al. [9], Betsch et al. [29]). We use the anger emotion as a proxy for dissatisfaction with the government. We also lag the variable.

3. Compliance: as a proxy for collective responsibility. We follow Sarracino et al. [34] and define compliance as the degree of association between people's behaviours and COVID-19 containment policies to construct the compliance variable. We use information gathered from Google Mobility Reports (the change in duration from the residential category) (Google [10] [11]) and the Stringency Index, which consists of the following nine indicators: school closing, workplace closing, events cancelled, restriction of gatherings, closed public transport, staying at home requirements, restrictions of internal movements, international travel controls, and public information campaigns. The Stringency Index ranges from 0 to 100, with 100 being the most stringent, and we sourced it from Oxford's COVID-19 Government Response Tracker (Hale [8]). Therefore, we estimate the following equation:

$$res_{ct} = \alpha + \beta_{ct} \cdot Country_c \cdot Day_t \cdot Policy_{ct} + \delta_c \cdot Country_c + \lambda_{ms} + \varepsilon_{ct} \qquad (1)$$

where $res_{ct}$ is residential mobility in country $c$ on day $t$; *Country* is a vector of dummies for each country included in the dataset; *Day* is a vector of dummies for the days from 1 February 2021 to 31 July 2021. We focus on this period because prior to February 2021, the vaccine rollout did not occur in all countries under investigation. $Policy_{ct}$ represents the stringency of containment policies in country $c$ on day $t$. A vector of dummies is depicted by $\lambda$ for each combination of month $m$ and hemisphere $s$, to account for the different seasons and evolution of the pandemic among the Northern and Southern hemispheres. The coefficient $\beta_{ct}$ is our measure of compliance. It provides the correlation between policy stringency and mobility by country and day. We are aware that creating a daily compliance measure risks introducing noise in the correlation. However, to fulfil our aim of determining the daily evolution of positive attitudes towards vaccines, we need to assess daily changes in compliance.

4. All tweets related to vaccines (Greyling et al. [9]): this is a proxy for the prominence of vaccines as a conversation topic.

5. Daily COVID-19 vaccine doses administered per million people (Mathieu et al. [13], Betsch et al. [29]): a proxy for how well a country handles the vaccine rollout. We lag this variable to address possible endogeneity that might spread from confounding factors. The rollout or lack thereof also proxies various constraints such as problems with the physical availability of the COVID-19 vaccine, lack of geographical accessibility, or signalling a less than adequate appeal for vaccination services uptake. We find that the VPAI and the daily vaccines have an inversely proportional relationship; therefore, we transformed this variable using a hyperbolic function.

6. Daily total new cases: a proxy for the evolution of the COVID-19 pandemic across all ten countries (Hale et al. [8]). In our models, we lag new cases to capture people's expectations of the trend of the pandemic.

7. Vaccine policy: we control for the vaccination policy across our ten countries. According to Hale et al. [8], a vaccination policy is classified as follows: 0—no vaccine available; 1 –

**Table 2. Descriptive statistics of the variables included in the estimations of attitudes against the COVID-19 vaccine.**

| Variable | Observations | Mean/ Frequency (%) | Std Dev. | Min | Max |
|---|---|---|---|---|---|
| VPAI | 1,780 | 0.35 | 0.12 | 0.10 | 0.91 |
| Lagged trust in the COVID-19 vaccine | 1,780 | 0.37 | 0.09 | 0.16 | 0.91 |
| Stringency index | 1,780 | 60.88 | 17.48 | 22.22 | 87.96 |
| Residential mobility | 1,780 | 8.32 | 24.62 | -29.67 | 50.85 |
| Lagged compliance | 1,780 | 1.07 | 0.273 | 0.621 | 2.37 |
| Lagged anger towards the government # Daily vaccinations | 1,780 | 0.84 | 0.54 | 0.00 | 2.52 |
| Vaccine tweets* | 1,780 | 106.32 | 108.20 | 6 | 690 |
| Lagged new daily vaccinations* | 1,780 | 231776.60 | 219928.20 | 0 | 873515 |
| Lagged new daily cases* | 1,780 | 148 | 154 | 0 | 701 |
| Vaccine policy | | | | | |
| 0 | 605 | 25.80 | - | - | - |
| 1 | 119 | 5.88 | - | - | - |
| 2 | 444 | 21.13 | - | - | - |
| 3 | 574 | 22.08 | - | - | - |
| 4 | 353 | 13.98 | - | - | - |
| 5 | 305 | 11.13 | - | - | - |

Source: Authors' calculations.

*Note: Vaccine tweets were logged, and the hyperbolic function of new daily vaccination was derived; the new daily cases were logged, and all variables were smoothed using a seven-day average.

vaccine available for one of the following groups: key workers / clinically vulnerable groups / elderly groups; 2—available for two of the abovementioned groups; 3—available for all the abovementioned groups; 4—available for all three groups plus partial additional availability (select broad groups/ages) and 5 –the vaccine is universally available.

8. To capture anger directed towards the government, we use an interaction variable 'government anger' interacted with 'new daily vaccinations'. This variable captures the anger expressed towards the government given the number of new vaccinations per day. We use the above as a proxy for people's dissatisfaction with the vaccination rollout, which also encapsulates procurement, capacity and corruption issues, and accessibility of the vaccines.

Table 2 provides summarised statistics for the variables included in our study.

## 3.3 Methodology

We first use descriptive statistics and graphs to analyse the trend in the VPAI over time and compare the results for the Northern and Southern hemispheres and across the ten countries in our sample. Our descriptive analysis includes topic modelling of the tweets per country. We explore the text corpus by applying NLP and statistical analysis. The main statistical procedure we use in the topic modelling is factor analysis. We attempt to uncover the text corpus' hidden thematic structure (topics) using this method. Secondly, we use various econometric techniques to derive and test the robustness of the relationships between our selected covariates and the attitudes towards vaccines.

The correlation between the VPAI and the covariates over time is likely to be affected by confounding factors, such as the severity of the pandemic, exposure to different types of social media, emotional well-being (depression) of the people, accessibility of the vaccine, the prejudice built into the social-cultural environment and the seasons of the year. Therefore, we resort

to various econometric techniques to address biases arising from the confounding effects of these variables.

Ideally, we would like to estimate the following equation:

$$VPAI_{ct} = \beta_0 + \beta_1 Vac\_Trust_{ct-1} + \beta_2 Gov\_Anger_{ct-1} + \beta_3 Compliance_{ct-1} + \beta_z X_{ct} + \lambda_m + \mu_c + \epsilon_{ct} \quad (2)$$

where $VPAI_{ct}$ is the vaccine positive attitude index as defined in section 3.2.1 for country $c$ on day $t$; $Vac\_Trust_{ct-1}$ (see section 3.3.1) is the average level of trust related to the COVID-19 vaccine for country $c$ on day $t-1$. $Gov\_Anger_{ct-1}$ is the average level of anger towards government for country $c$ on day $t-1$; $Compliance_{ct-1}$ is the average level of compliance as defined in section 3.2.2 for country $c$ on day $t-1$ $X_{ct}$ is a vector of variables, $\lambda_m$ are month effects capturing common effects across countries, such as seasonal effects (changes in seasons), the evolution of the pandemic and holiday seasons (July for the Northern hemisphere means Summer holidays and for the Southern hemisphere Winter holidays), while $\mu_c$ are country effects.

**3.3.1 Pooled ordinary least squares (POLS).**   As a baseline model, we use a POLS estimation. To address the bias that might spread from reverse causality, we lag 'trust in the COVID-19 vaccines', 'anger towards government', 'compliance', 'the daily number of COVID-19 vaccinations' and 'cases'. To address heteroscedasticity, we use robust standard errors in the estimated models.

**3.3.2 Fixed effect (FE) estimation.**   Having the benefit of a panel dataset allows us to control for additional biases, particularly unobserved confounding factors. Specifically, the FE approach reduces the impact of confounding by time-invariant factors, such as the unobserved and, in this instance, observed characteristics of the countries.

We use the Haussmann test to test if the FE model rather than the Random Effects (RE) model is the most efficient estimator in the current study. We reject the null hypothesis that there is "*no correlation between the unique errors and the regressors in the model*", confirming that the FE will give the most robust estimations.

The country (individual) FE included in the model addresses the unobserved time-invariant heterogeneity between countries, which considerably reduces the risk of the confounding factors discussed above. Additionally, the FE model partly addresses bias originating from omitted observed variables (related to country characteristics). However, the FE model cannot address bias for unmeasured time-varying confounding factors or reverse causality. To further address reverse causality, we turn to Instrumental Variable regressions.

**3.3.3 Instrumental variable (IV) regression.**   In addition to the lagged variables introduced in the POLS and the FE estimations, we also use an IV model to address possible endogeneity and reverse causality. We use the Generalised Method of Moments (GMM) estimation rather than the Two-Stage Least Square (2SLS) estimator, due to the efficiency gains derived from using the optimal weighting matrix. The efficient GMM estimator is robust to heteroscedasticity of unknown form.

We instrument 'lagged trust in the COVID-19 vaccine' and 'lagged compliance', with 'lagged fear of the vaccines', 'lagged disgust with the vaccines' and a two-day lag in 'compliance'. We use the Hansen's J statistic to test for over-identifying restrictions. The joint null hypothesis is that the excluded instruments are valid instruments, i.e. uncorrelated with the error term and correctly excluded from the estimated equation. A rejection casts doubt on the validity of the instruments. However, in our specified model, serial correlation is present as the error term in one period is correlated with the errors in previous periods. This causes the estimated variances of the regression coefficients to be biased, leading to unreliable hypothesis testing. Therefore, we consider the IV estimations with the POLS and FE estimations.

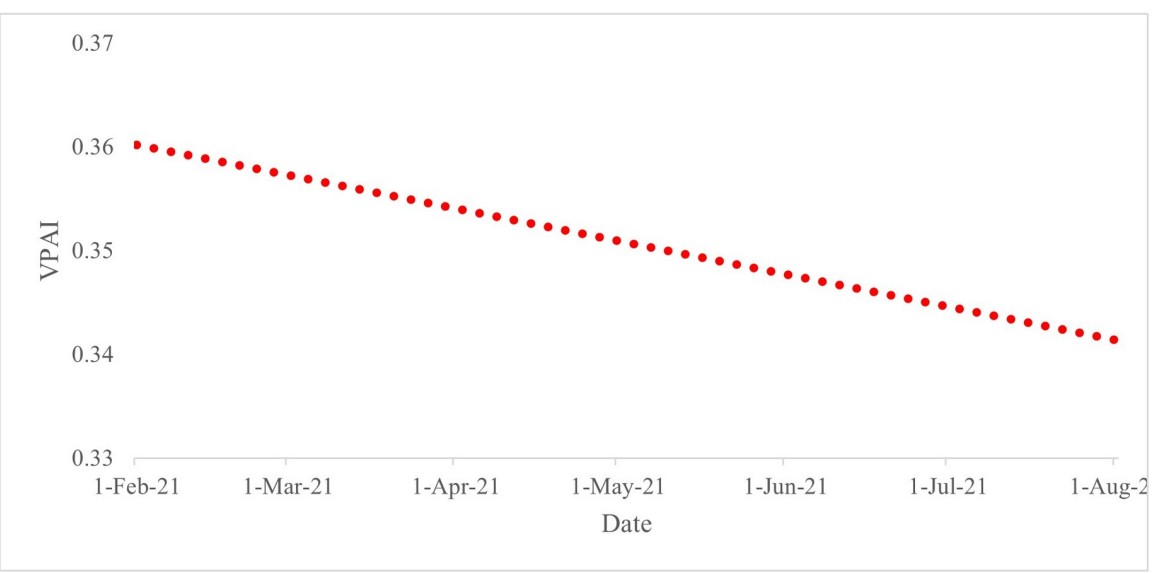

**Fig 4. Trend in positive attitude from February 2021 to the end of July 2021 for the whole sample.** Source: Authors' calculations.

# 4. Results

## 4.1 Results of the trend in the VPAI

We first focus on our descriptive analysis (graphs) to explain the trends in the VPAI towards the COVID-19 vaccines for the period 1 February 2021 to 31 July 2021. We describe the trends in our overall sample, the different hemispheres and lastly, each country. In all instances, we report the findings on the VPAI using NRC, and as a robustness test, we repeat the analyses using Sentiment140. We report the results using Sentiment140 in the supporting information section.

**4.1.1 Trends in the VPAI.** When we consider Fig 4, we see that the trend in the VPAI towards the COVID-19 vaccines across all countries is downwards; we note an almost 8 per cent decrease over time. Section 4.1.2 discusses possible explanations for this downward trend.

Additionally, we note from Fig 5 that the downward trend in positive attitude holds across both the Northern and Southern hemispheres. However, the downward trend seems stronger in the Southern hemisphere than in the Northern Hemisphere.

**4.1.2 Trends in positive attitude per country.** If we consider the individual countries, Fig 6 shows the trend in the VPAI towards the COVID-19 vaccines for each of the ten countries and indicates that the attitude improved in only two countries, namely Belgium and the Netherlands. For the remaining eight countries, the trend was negative over time.

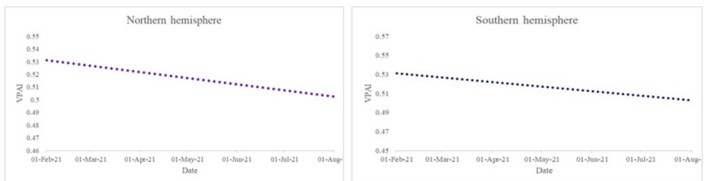

**Fig 5. Trend in positive attitude across the Northern and Southern hemispheres from February 2021 to the end of July 2021.** Source: Authors' calculations.

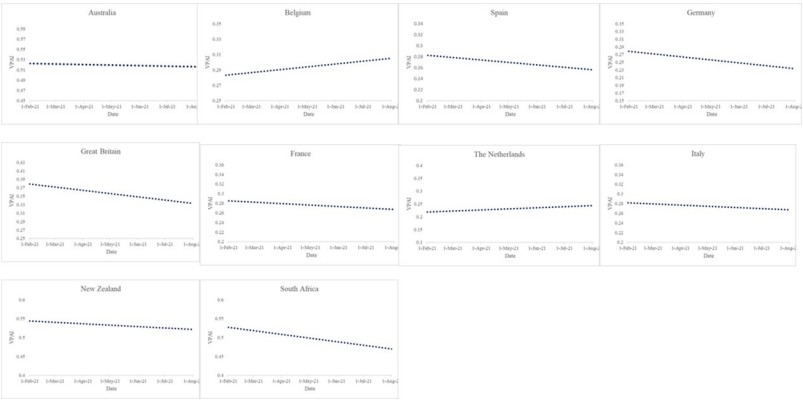

**Fig 6. Trend in the positive attitude for each of the ten countries.** Source: Authors' calculations.

To explain the trends in the VPAI for our individual countries, we relied on existing literature and our topic modelling.

Upon further investigation into Belgium, we found that the positive trend in the VPAI in Belgium was likely due to the steps taken to correct government failure that plagued the country in 2020. In 2020 (Villani et al. [35]), Belgium was the European country with the highest loss of life and hospitalisation rate relative to the size of the population in Europe. According to Vanham [36], Belgium was also hit with capacity issues, struggling to get vaccination centres up and running because of vaccine delivery delays. It seems that the Belgian people did not trust information coming from their government after reports of political favouritism in deciding who would get what little vaccine stock was available were leaked (Vanham [36]). The above events led to widespread anger towards the politicians for making COVID-19 a political game.

However, the government took many steps to correct the situation, likely turning the attitude towards the COVID-19 vaccine positive. The Belgium government set up a COVID-19 task force responsible for addressing logistics and capacity issues. According to Vanham [36], the high uptake of vaccinations could also result from lockdown regulation policies being relaxed during spring, which would depend on the vaccination rates rather than case numbers or hospitalisation rates. People wanting to return to 'normal' reacted positively to the policy. Our topic modelling also revealed that Belgians was found to be optimistic about the effectiveness of the COVID-19 vaccines, including AstraZeneca. However, they were pleased that younger people could receive the Janssen vaccine.

The Netherlands was the last European country to start their vaccine rollout on 6 January 2021. Their rollout was hampered by a poor vaccination policy and a conservative strategy that kept more than 40 per cent of its vaccines from being used (Bahceli [37]). Additionally, during the beginning phase of the vaccine rollout, young Dutch adults (18 to 34 years, which constitutes approximately 25 per cent of the total population) who were willing to receive the COVID-19 vaccine constantly lagged about ten percentage points after the average percentage of the whole population (Vollmann and Salewski [38]). To encourage a positive attitude towards vaccines, the Dutch government spent around €6 million (at the time of the study) on information campaigns to increase the vaccine uptake by informing the public about the safety of the various COVID-19 vaccines (Bahceli [37]). Vollmann and Salewskis' [38] results show that the relationship between information campaigns and a positive attitude towards vaccines leads to higher vaccine uptake. The Netherland's fully vaccinated population increased

significantly from 2,430 people on 31 January 2021 to 9,288,187 by 1 August 2021 (Mathieu et al. [13]). We also found (from topic modelling) that the Dutch believed the benefits of getting the COVID-19 vaccine outweighs any potential costs, which is why they expressed support for the elderly and the vulnerable to be prioritised and getting the younger generation vaccinated as soon as possible.

Literature and topic modelling revealed the following likely explanations for those countries that experienced downward trends in positive attitudes over time.

Australia follows a Federal system of government, and contradictory government-implemented regulations across the different states led to widespread confusion regarding COVID-19 vaccines and caused a downtrend in attitudes (Attwell et al. [39]). This was confirmed by our topic modelling as well. New South Wales (NSW) (Greater Sydney region), home to 32.33 per cent of the total Australian population, was at the time of writing this paper in lockdown, while other regions were not regardless of case numbers. These discrepancies in COVID-19 regulations resulted in a significant proportion of people living within NSW refusing to comply with government-imposed regulations. Topic modelling illustrated that there was anger over government-mandated lockdowns, a perceived incompetent government and fear of side effects. In July 2021, approximately 15,000 people (most not wearing masks) protested against the lockdowns, and they demanded their liberties be restored (Swain [40]). Fig 6 shows the downward trend in positive attitudes towards the COVID-19 vaccine.

Spain shows a downward trend in the VPAI. A possible explanation could be that the government delayed action and did not have workable contingency plans in place. This led to a failure containing the virus early on, nearly overwhelming the health system and causing Spain to experience one of the highest death rates attributable to COVID-19 (Casasnovas et al. [41]). This lack of action caused that during the first five months of 2021, Spain's COVID-19 vaccination campaign progressed slowly and failed to reach marginalised populations (Lazarus et al. [42]). In mid-April, when 13 per cent of Great Britain's citizens were fully vaccinated, only about 7 per cent of Spaniards were similarly protected (Mathieu et al. [13]). The above rhetoric was echoed through our topic modelling, where Spaniards expressed anger towards their government for failing to rollout the COVID-19 vaccine efficiently while at the same time expressing fear for the unknown long-term side effects of the vaccine.

According to Sprengholz et al. [43], the Germans responded with anger towards their government's proposed policy to contain COVID-19. The policy stated that only vaccinated people would be allowed to enter venues like sports stadiums, movie theatres or restaurants because they deemed the residual risk high in such places. In July 2021, Chief of Staff Helge Braun announced that he did not expect another COVID-19 related lockdown in Germany (Schultheis [44]). However, this would mean that if there were a future outbreak, the liberties of the unvaccinated would be taken away with immediate effect. Additionally, according to Mario Czaja, head of the Berlin Red Cross, Germany has seen an increase in people not showing up for their vaccination appointments, with 5 to 10 per cent missing appointments daily since July 2021 (Reuters [45]).

The results of the topic modelling of tweets showed that the Germans were angry as they had to wait in queues at the vaccine centres. Furthermore, they were unhappy to receive AstraZeneca due to the news of potential blood clotting. They also mentioned concerns about side effects, but many tweets reassured that these were only mild. Lastly, Germany's anti-lockdown movement, the Querdenken, has been very active in spreading conspiracy theories ranging from the idea that "masks are deadly" to "vaccines can alter your DNA" (BBC Trending [46]). These could all have contributed to the decreasing trend in the VPAI.

In Great Britain, the government faced criticism because of their vaccination policy, which at the time of writing this paper, was yet to approve the COVID-19 vaccine for 12–15-year-

olds (Mason and Elgot [47]). This meant sending children back to schools with inadequate mitigations for COVID-19 in place, which could lead to widespread infections and more disruptions to learning. Additionally, trust in the ability of the government to see this pandemic through has decreased since the announcement of their so-called 'Freedom Day' (Donovan [48]). Freedom Day brought with it a lifting of any remaining COVID-19 restrictions and came amidst 47,000 new cases of COVID-19 being reported in the previous 24 hours (Donovan [48]). The decision of Freedom Day brought with it 1,200 scientists worldwide criticising the decision to open up, saying it could pose a threat to the entire world if daily cases increased exponentially and vaccine-resistant mutations of the virus were allowed to develop (Ball [49]). In our topic modelling, we found that people were concerned about Freedom day and not wearing masks or applying social distancing rules. They were also concerned about increasing numbers of COVID-19 cases and the blood clotting side effect from AstraZeneca. The downward trend in the VPAI is likely (Fig 6) a product of all the accumulated issues.

In France, introducing a stringent vaccination policy known as 'COVID-19 vaccine passports' has decreased the positive attitude towards the COVID-19 vaccine (The Economist [50]). The news that movie theatres, museums and sports venues have begun asking visitors to provide proof of a COVID-19 vaccination or a negative test has many French nationals angry but willing to take the vaccine simply to return to their once normal way of living (The Economist [50]). The topic modelling of tweets supported the above mentioned and highlighted the reluctance to accept the health pass and scan QR codes. They were also concerned about the side effects and mentioned that the "Covidliste" has a long waiting list. "Covidliste" is a voluntary and civic initiative allowing the connection between vaccination volunteers and health professionals who have vaccine doses.

Italy, the second-worst Northern hemisphere country concerning people not fully vaccinated (52 per cent at the time of writing the paper), has faced an uphill battle to increase the COVID-19 vaccine uptake since the start of their vaccine rollout on 27 December 2020 (Roberts [51]). They decided to take a tough stance, approving emergency legislation to make COVID-19 vaccines mandatory for all healthcare workers, including pharmacy staff (Roberts [51]). Individuals working in this industry who refused the COVID-19 vaccine would be transferred to another job or suspended without pay for up to a year. In addition, they introduced vaccine passports. From both the topic modelling and Paterlini [52], we saw that Italians demonstrated and protested against Italy's use of these passports and the subsequent green passes.

Furthermore, the emergency legislation faced fierce resistance from Italy's deeply rooted anti-vaccine movement, which has been fostered in part by populist political forces (Roberts [51]). These included the 5 Star Movement, which entered government in 2018 promoting vaccine hesitancy. Public trust in the vaccine has also taken a hit after the country temporarily decided to suspend the use of the Oxford/AstraZeneca vaccine after several deaths (Roberts [51]).

In New Zealand, the Aged Care Association (Wallis [53]) described the COVID-19 rollout as a 'shambles'. This is due to the government of the day being responsible for a slow rollout of the vaccine because they and the country as a whole became complacent (Vance [54], Thaker and Floyd [55]). At the time of writing this paper, New Zealand found itself in a level-4 lockdown (the most stringent level of lockdown), even though it did not have a positive COVID-19 case during the previous six months. New Zealand's zero COVID-19 strategies were successful until the first Delta-variant positive case was announced. Soon, the government realised that they did not have enough vaccines to vaccinate everyone fully, as previously promised. This was partly because of the government's strategy early in 2021 to reject the cheaper but potentially less effective vaccines like those made by AstraZeneca in favour of the high-performing vaccine made by Pfizer/BioNTech (Satherley [56]).

Additionally, the increased spread of misinformation about the COVID-19 vaccine has increased distrust towards the vaccine (Thaker and Subramanian [57]). Unfortunately, for New Zealand, the spread of misinformation and conspiracy theories are rampant among its ethnic minority and most vulnerable community (Tukuitonga [58]). The lack of trust in the government after failing in their vaccine rollout by not reaching the most marginalised first (topic modelling) is precisely what was needed for the spread of misinformation to take hold. Our results confirm the study done by Paul et al. [19], who found that distrustful attitudes towards vaccination were higher amongst individuals from ethnic minority backgrounds which isn't a surprise since many minority groups have good reason not to trust the government given their historical mistreatment. All of the abovementioned likely contributed to the downtrend in the VPAI.

South Africa' has faced problems such as capacity issues, mistrust in the government and anti-vaccination campaigns (Cocks [59]), which contributed to the decrease in a positive attitude towards the COVID-19 vaccines (see Fig 6). From as early as December 2020, it seemed that the COVID-19 strategy was haphazard apart from its dependency on their fragile COVAX arrangement. After receiving their first delivery of the AstraZeneca vaccine on 1 February 2021, it seemed that the government also did not have a clear vaccination policy (van den Heever et al. [60]). The Health Minister created confusion in the public arena when he announced that the AstraZeneca vaccine did not demonstrate efficacy against mild to moderate COVID-19 and placed the rollout of the vaccine on hold. The announcement by the Health Minister caused a decrease in trust in the COVID-19 vaccine and likely contributed to the downward trend in the VPAI. Local scientists criticised the decision, and the World Health Organisation did not support it (van den Heever et al. [60]). This decision left approximately 17 million high-risk population unvaccinated (van den Heever et al. [60]). During the winter months from June to September 2021, South Africa lost 25,660 lives to COVID-19 (Roser et al. [12]). According to van den Heever et al. [60], this probably could have been avoided if the South African government had not been plagued by corruption and mismanagement during its response to the pandemic. By August 2021, South Africa saw 'vaccine apathy' or 'vaccine fatigue', with the number of people coming forward to be vaccinated dropping below 200,000 a day, falling short of the set target of 300,000. According to a study conducted by the Human Sciences Research Council and the University of Johannesburg (Cooper et al. [61]), the vaccine-hesitant cite three primary concerns, which could contribute to the downtrend in positive attitudes: side effects, effectiveness, and distrust of the vaccine and institutions.

To summarise, the downward trend in positive attitudes is partly due to a fear of the side effects, but many other factors also contribute. These include dissatisfaction with governments' rollout plan, procurement, corruption, resistance to mandatory vaccination and the use of COVID-19 passports.

## 4.2 POLS, FE and IV regression results

This section discusses the results of those covariates that are significantly related to the VPAI and, therefore, when addressed, could improve attitudes and the uptake of COVID-19 vaccines.

In Table 3, the results of the POLS estimation controlling for month and country fixed effects are similar to the results of the FE model and the IV regression. The covariate 'trust in the COVID-19 vaccine' is statistically significant and positively related to the VPAI across all the estimated models. The methods have different advantages; thus, the joint results confirm that trust in the COVID-19 vaccine is a robust correlate of VPAI. We assume that when trust in the vaccine increases, the fear of adverse side effects decreases and that the positive attitude

**Table 3. Results from POLS with FE and IV.**

| Variable | POLS | | FE | | IV | |
|---|---|---|---|---|---|---|
| VPAI | Coefficient | SE | Coefficient | SE | Coefficient | SE |
| Lagged trust in the COVID-19 vaccine | 0.2938*** | (0.0281) | 0.2938*** | (0.0193) | 0.3127*** | (0.0999) |
| Lagged compliance | -0.0176*** | (0.0060) | -0.0176*** | (0.0037) | -0.0170*** | (0.0064) |
| Lagged anger towards the government # Daily vaccinations | -0.0274*** | (0.0073) | -0.0274*** | (0.0073) | -0.0273*** | (0.0072) |
| Lagged new daily vaccinations | 0.0055*** | (0.0013) | 0.0055*** | (0.0014) | 0.0055*** | (0.0014) |
| Lagged new daily cases | -0.0032*** | (0.0080) | -0.0032*** | (0.0011) | -0.00324*** | (0.0080) |
| Log vacc tweets | -0.0049* | (0.0029) | -0.0049* | (0.0029) | -0.0047* | (0.0029) |
| Vaccine policy (Reference—Level 0) | | | | | | |
| Level 1 | 0.0598*** | -0.0125 | 0.0598** | -0.0232 | 0.0596*** | (0.0123) |
| Level 2 | 0.0451*** | -0.0091 | 0.0451** | -0.0227 | 0.0454*** | (0.0105) |
| Level 3 | 0.0476*** | -0.0088 | 0.0476** | -0.0228 | 0.0480*** | (0.0092) |
| Level 4 | 0.0536*** | -0.0092 | 0.0536** | -0.0227 | 0.0542*** | (0.0103) |
| Level 5 | 0.0473*** | -0.0102 | 0.0473** | -0.0233 | 0.0477*** | (0.0109) |
| Country FE | Yes | | Yes | | Yes | |
| Month FE | Yes | | Yes | | Yes | |
| N | 1727 | | 1727 | | 1727 | |
| Adjusted $R^2$ | 0.867 | | 0.422 | | 0.866 | |
| Hansen J-Statistic of overidentification | | | | | p = 0.6544 | |

Source: Authors' calculations.

Robust Standard errors in parentheses

* $p < 0.10$,

** $p < 0.05$,

*** $p < 0.01$

towards vaccines improves. This finding is in line with the works done by Akarsu et al. [14], Fisher et al. [15], Freeman et al. [16], Ward et al. [17], Seale et al. [18]), Paul et al. [19], Sallam [20] (refer to section 2).

Compliance, the act of complying with government-mandated regulations to curb the spread of COVID-19, is statistically significant and negatively related to the VPAI. We interpret this as when people are reluctant unwilling (do not feel like) to comply with orders such as mask-wearing, staying at home etc., then those individuals would be more willing to receive the COVID-19 vaccine, hence a more positive attitude (thus an inverse relationship). A study conducted by Wright et al. [62] investigated the relationship between vaccinated individuals' willingness to comply and the implemented behavioural regulations. The entire premise of the study is that vaccinated individuals believe they are less at risk because of their vaccination status. People think when vaccinated, they do not need to comply with, for example, mask-wearing, social distancing etc., therefore creating a more positive attitude towards vaccines. This finding is informative to policymakers as a message of "less strict regulations" after vaccination can increase vaccine uptake.

The variable 'anger towards the government' interacted with the new daily vaccinations is statistically significant and negatively related to the VPAI. Therefore, when people's dissatisfaction with the government increase, positive attitudes decrease. Analysing the tweets, we find that people are angry with governments due to the lack of procurement, procurement of the incorrect vaccine, the rollout of the vaccination plan and corruption within governments. This anger directed at governments due to a lack of access to vaccines sabotages the positive attitude towards vaccines and hinders the uptake of vaccines.

The relationship between the VPAI and the number of daily new vaccinations is inversely proportional, significant and positively related. This implies that when the daily number of vaccines administered are very low, the positive attitude is high, but as the number of vaccines administered per day increases, the positive attitude starts to plateau. Also, using the vaccine rollout or the lack thereof, as a proxy for constraints in information campaigns, the physical availability of vaccines or a lack of geographical accessibility, we can see how important it is to overcome any barriers which might impede the intention to be vaccinated (Cylus and Papanicolas [63]).

We find that daily new cases, a proxy for the evolution of the COVID-19 pandemic across all ten countries, is statistically significant and negative. If the daily cases are high, the positive attitude towards the vaccine is relatively low, but as the daily cases start decreasing, the positive attitude improves.

Controlling for the vaccination policy, thus the groups that can access the COVID-19 vaccine, we find that when more groups of people can access the vaccine, for example, all age groups compared to fewer groups, it is positively related to the VPAI. Once again, showing that when more people have access to the COVID-19 vaccine, positivity towards the vaccine is enhanced.

The number of vaccine-related tweets is statistically significant and negatively related to the VPAI in all the estimated models. This implies that, as the number of tweets related to vaccines increases, the positive attitude towards vaccines decreases. We found that the number of vaccine-related tweets increased over time; thus, vaccines became a "hot" topic of discussion. Topic modelling on the vaccine-related tweets indicates that negative sentiment is related to, among others: the long-term effects of the COVID-19 vaccine; dissatisfaction with vaccine passports and QR code scanning; concerns about the Delta variant; anger about procurement; struggling to make appointments (vaccine rollout) and conspiracy theories. People also express their discontent with social distancing and wearing masks. Our topic modelling results are in line with studies such as Küçükali et al. [23], Nuzhath et al. [24], Bonnevie et al. [2] and Thelwall et al. [25].

All of the above leads us to believe that increased tweets with negative connotations to the COVID-19 vaccine decrease positive attitudes towards vaccines.

In summary, we see that those variables that can improve the positive attitude towards vaccines are related to information about the safety and side effects of the vaccines (increased trust in vaccines) and a balance between the strictness of regulations and access to vaccines. Additionally, increased trust in the governments' capabilities, honesty of governments and dealing with capacity constraints can decrease the dissatisfaction with governments and increase vaccine uptake. Precise information about the COVID-19 vaccines in general also disseminated via social media can increase positivity towards the COVID-19 vaccine. Misinformation about COVID-19 vaccines and social media should be monitored, and campaigns against this misinformation should be launched. Vaccines should also be made accessible to all groups of people.

## 5. Conclusions

In this study, we constructed a real-time Vaccine Positive Attitude Index (VPAI) derived from Big Data to illustrate the evolution of people's positive attitudes toward the COVID-19 vaccine across ten countries. Our descriptive analysis showed that the VPAI generally indicates a decline in attitude over the time period investigated. When we consider the different hemispheres, the trend is downwards in the Northern and Southern hemispheres. Furthermore, considering the ten individual countries, only Belgium and the Netherlands experienced a positive trend in the VPAI, whereas the other countries experienced a negative trend.

Using POLS, FE and IV regression models, we determined which variables are significantly related to the VPAI and, therefore, could increase the uptake of COVID-19 vaccines if addressed by policy measures. We found that those variables that could improve people's attitudes towards vaccines were: i) information related to the safety and side effects of the vaccines, ii) increased trust in governments in conducting the vaccine rollout and handling procurement and capacity issues, iii) cognisance of the compliance versus the vaccine up-take decision, and iv) better information about the COVID-19 vaccines in general, but primarily disseminated via social media.

These results give policymakers the necessary information to increase positive attitudes towards the COVID-19 vaccine. Policymakers should focus on improving trust in the COVID-19 vaccines. They could more openly disseminate information regarding the vaccine, do it in layman's terms, and acknowledge people's fears, anger, and other negative emotions. They can emphasise the stringent safety and efficacy standards of the COVID-19 vaccine development process, thus fostering trust in the vaccine. All of this may increase vaccine confidence. Additionally, countries can overcome the lack of accessibility to vaccination clinics by following examples set by the United States of America to introduce mobile vaccination clinics to reach people in remote areas.

Furthermore, policymakers should implement policies to increase people's sense of collective responsibility. This can be done by raising awareness of emotional manipulations by anti-vaccine disinformation efforts and activating positive emotions such as altruism and hope as part of vaccine education endeavours. Another potential strategy is to elicit positive emotions toward helping one's community restore health and well-being when deciding to vaccinate against what is called the most consequential disease of our time.

Lastly, the study has limitations. We only used Twitter in our analyses as other social media platforms, for example, Facebook, is highly protected. In the future, additional social media platforms should be analysed. A drawback of social media data, including Twitter, consists in their lack of representativeness of populations. However, a case can be made that the sample sizes using Big Data contributes to robust results. When translating the tweets from a non-English language, errors can occur at three levels: lexical, syntactical, and discursive. These errors inevitably can cause unintelligibility which means that the tweet will be disregarded. However, in the current study, we used lexicons in the original languages (if available) and compared the results to the translated text corpus. We find the time-series trends using the original language and translated text well correlated.

Furthermore, we only examined the tweets for a specific time period. Therefore, we cannot examine the tweet's effects following the study. Another limitation is the timeline of research and the publication thereof. Publication rates cannot keep up with the tempo at which the pandemic evolves; therefore, by the time of publication, new COVID-19 variants might have emerged, and research on vaccines would have progressed. Nonetheless, the knowledge gained in this research contributes to more informed decision making in the future. Lastly, the countries used for this study are solely based on data availability, which can be considered a limitation, especially given a non-characterisation of how other countries could present attitudes towards vaccines. In future studies, with more resources, we can expand the text corpus to include more countries over a longer period with more sources of Big Data.

## Supporting information

**S1 Fig. Trend in positive attitude from February 2021 to the end of July 2021 for the whole sample, using Sentiment140.** Source: Authors' calculations.
(TIF)

**S2 Fig. Trend in positive attitude from February 2021 to the end of July 2021 per hemisphere, using Sentiment140.** Source: Authors' calculations.
(TIF)

**S3 Fig. Trend in positive attitude from February 2021 to the end of July 2021 per individual country, using Sentiment140.** Source: Authors' calculations.
(TIF)

**S4 Fig. Trend in positive attitude including false negatives related to governments (VPAI2) for the whole sample from February 2021 to the end of July 2021.** Source: Authors' calculations.
(TIF)

**S5 Fig. Trend in positive attitude including false negatives related to governments (VPAI2) for the different hemispheres from February 2021 to the end of July 2021.** Source: Authors' calculations.
(TIF)

**S6 Fig. Trend in positive attitude including false negatives related to governments (VPAI2) for the individual countries from February 2021 to the end of July 2021.** Source: Authors' calculations.
(TIF)

**S1 File.**
(DTA)

## Author Contributions

**Conceptualization:** Talita Greyling, Stephanié Rossouw.

**Data curation:** Talita Greyling, Stephanié Rossouw.

**Formal analysis:** Talita Greyling, Stephanié Rossouw.

**Investigation:** Talita Greyling, Stephanié Rossouw.

**Methodology:** Talita Greyling, Stephanié Rossouw.

**Project administration:** Talita Greyling, Stephanié Rossouw.

**Resources:** Talita Greyling, Stephanié Rossouw.

**Software:** Talita Greyling, Stephanié Rossouw.

**Supervision:** Talita Greyling, Stephanié Rossouw.

**Validation:** Talita Greyling, Stephanié Rossouw.

**Visualization:** Talita Greyling, Stephanié Rossouw.

**Writing – original draft:** Talita Greyling, Stephanié Rossouw.

**Writing – review & editing:** Stephanié Rossouw.

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
