## [Decision Letter · Decision Letter 0]

7 Jan 2022

PONE-D-21-34544Positive attitudes towards COVID-19 vaccines: A cross-country analysis.PLOS ONE

Dear Dr. Rossouw,

Thank you for submitting your manuscript to PLOS ONE. After careful consideration, we feel that it has merit but does not fully meet PLOS ONE’s publication criteria as it currently stands. Therefore, we invite you to submit a revised version of the manuscript that addresses the points raised during the review process.

We look forward to receiving your revised manuscript.

Kind regards,

Prof. Anat Gesser-Edelsburg, Ph.D.

Academic Editor

PLOS ONE

Journal Requirements:

Reviewers' comments:

Reviewer's Responses to Questions

**Comments to the Author**

1. Is the manuscript technically sound, and do the data support the conclusions?

Reviewer #1: Yes

Reviewer #2: Partly

Reviewer #3: Yes

2. Has the statistical analysis been performed appropriately and rigorously? 

Reviewer #1: Yes

Reviewer #2: I Don't Know

Reviewer #3: Yes

3. Have the authors made all data underlying the findings in their manuscript fully available?

Reviewer #1: Yes

Reviewer #2: Yes

Reviewer #3: Yes

4. Is the manuscript presented in an intelligible fashion and written in standard English?

Reviewer #1: Yes

Reviewer #2: Yes

Reviewer #3: Yes

5. Review Comments to the Author

Reviewer #1: The article uses Big Data to study a very important topic of COVID-19 vaccine hesitancy and factors that may enhance vaccine uptake. The authors provide an expansive picture of commonalities and diversities among varied countries, which contribute to the understanding of this vital issue. The article is well-written, maintaining clarity and logical flow in the description of the findings and their potential value.

Nonetheless, there are several elements that need to be improved in order for the article to be appropriate for publication, as follows:

1. The authors ‘mix’ between results and discussion. See section 4.1.2. the results should be presented as found, without the authors trying to provide an explanation to why each element was such found. These explanations (for example see lines 375-388 concerning Belgium or lines 389-396 concerning Netherlands) are not appropriate in the results section.

2. The authors should refrain from any judgmental comments (for example lines 395-396 – “It is safe to say that the information campaigns have paid off”). Scientific writing should state possible explanations to any phenomenon that is identified, usually based on previous publications or new theoretical hypotheses, but without any judgment of any kind (neither positive or negative). Furthermore, the authors have not shown any support that this is actually the case and that the information campaigns were the cause for the results identified; more cautious writing is recommended. The same relates to sentences such as line 467 “South Africa's woes are almost too many to count”. These types of phrases should be revised so that the sentence presents the finding as is and not referred to in ‘literary-type’ descriptions.

3. Whenever any assumption (or claimed ‘fact’) is written – it should be based on previously published data from scientific sources. Sentences such as “Australians have a deep-seated mistrust in their government officials, who seem to implement contradictory regulations across different states” (lines 398-399) have to be supported by such publications.

4. Lines 505-507 – “Compliance, the act of complying with government-mandated regulations to curb the spread of COVID506 19, is statistically significant and negatively related to the VPAI. Unwillingness to comply likely 507 motivates people to get vaccinated. “ is not clear. I would assume that compliance would follow positive attitudes (towards the vaccine or towards the authorized agencies). How do the authors explain the negative correlation?

5. The authors at times make claims that they do not show any support for based on their data. For example, their claim in lines 534-537 “This implies that, as the tweets related to vaccines increase, the attitude towards vaccines decreases. This may likely be because many of the tweets contain misinformation or conspiracy theories rather than campaigns and information to encourage being vaccinated - and, therefore, decreases positive attitudes towards vaccines.”. To support such a claim, they should check whether the tweets they investigated actually show this tendency.

Reviewer #2: The results reported in the paper are interesting as such, since the topic of public reaction to the COVID-19 pandemic and the related interventions is a timely one. However, to this referee it seems that the descriptive interest of the results dominates over the analytical one, due to various technical issues, as detailed below.

Why do authors call the analysis "panel analysis" when there is no panel of individuals followed at more than one time-point? Considering a country as an individual is a complicated assumption... In reality, the data is a sequence of cross-sectional analyses within each country without control of the relative media-related weight or publication frequency of pro- and anti-vax individuals in the data stream and therefore maybe biased as an estimate of the country specific sentiment...

The analysis methods, common in econometric panel data analyses, have been devised for individual data and, as also commented upon above, have to be motivated for use on aggregated, non-individual, data...

One could also suspect that people are more eager to announce an anti-vaccine stance than a neutral or pro-vaccine stance on social media, especially since the anti-vaxers are minorities, which therefore need more cohesion and solidarity...

The paper also seems to conclude that a part of the negative opinions are more due to vaccine availability and related problems than to an "ideological" anti-vaccine stance. A clearer separation of these issues in the analysis, in addition to the VPAI2 index, would help the understanding of vaccine-related sentiment...

In section 3.2.2 (Selection of covariates), many covariates are extracted from the same source as the response variable (the Tweet streams). Is there not a risk that this automatically induces correlations?

In the equation (2, p13), one wonders what common monthly effect (lambda_m) is expected in different hemispheres, in countries in different phases of epidemic spread...

The analysis in the paper is also to a large extent outdated, since so much , both in terms of restrictions and vaccinations, has happened since August '21...

In case of revision, the authors should incorporate some recent material, either in the analysis or the discussion.

Finally, regarding section 5 (Conclusions), a clearer distinction between conclusions strictly based on the analysis and authors opinions or suggestions for action should be made.

Small language errors or typos to be corrected, e.g. "illicit" instead of "elicit" (p2 l36) or "By analysing, the latter..." (p7 l162) instead of "By analysing the latter..."

This referee recommends a thorough revision and actualization of the paper before considering its possible publication.

Reviewer #3: Manuscript Identification

PONE-D-21-34544: This paper has two main objectives: 1) to conduct a cross-country panel analysis to investigate the trend in positive attitudes towards COVID-19 vaccines over time; and 2) lies with determining those variables which are significantly related to a positive vaccine attitude and can inform policymakers. The article has a good description of how previous studies analysed the emotions in vaccine-related tweets, and following this description, the authors present a considerable justification for why do this study, which is classified as the first study to use Big Data to determine the sentiment and emotions related to COVID-19 vaccines through a vaccine positive attitude index. The time series analysis is a good way to perform a point of view of this situation.

The authors' proposal is to present an analysis based on tweets from 10 countries, also dividing the analysis into countries in the southern hemisphere and countries in the northern hemisphere. This choice is based on data availability, which can be considered a limitation of the study, in view of a non-characterization of how other countries could present differences regarding the information provided.

Minor concerns

The text of the paper needs to be revised as it has some inconsistencies regarding the information.

1. About Figure 1 - it is necessary to modify the dates of the years presented in this figure, considering that the data were collected in 2021, the years change according to the countries. In addition, I also suggest a review of the percentages presented in terms of the number of people fully vaccinated, which differ from Table 1.

2. Section 3.2.1: the authors present some illustrations based on word clouds generated by tweets and inform that this was done for all country but in the paper, it's possible to find just 2-word clouds showing information by Great Britain and South Africa. It's important to inform why just these 2 countries were present in this analysis.

3. Section 3.2.2: Scale 5c. In the second paragraph, the word complacency is not present.

4. The authors do not present the limitations of the study, an important point for future studies on the subject. In view of some methodological characteristics, my suggestion is to highlight some points such as: the translation of tweets in a non-English language; choice of countries based on data convenience and availability.

5. There is no specific topic in the study for discussing the data, however, in the description of the results, the authors present some references that can be considered part of this discussion of the data. As this is an analytical study, it would be important to have a more in-depth discussion, comparing the results to other previous studies.

The manuscript has good methodological quality, is free of bias, needs some adjusts but the results are discussed based on the theoretical background properly of the manuscript theme.

So, the conclusions answered the aims of the study focused on the references and results.

The limitations of the study were not presented, and I suggest that this could be done.

6. PLOS authors have the option to publish the peer review history of their article (what does this mean?). If published, this will include your full peer review and any attached files.

Reviewer #1: No

Reviewer #2: No

Reviewer #3: **Yes: **Leonardo Pestillo de Oliveira

---

## [Author Response · Author response to Decision Letter 0]

14 Feb 2022

RESPONSE TO REVIEWERS

Manuscript ID PONE-D-21-34544

Positive attitudes towards COVID-19 vaccines: A cross-country analysis.

We thank the reviewers for the comments. Each has been comprehensively addressed, as set out below (authors' response shown in italics). Throughout the paper, we have also made improvements to enhance its quality and contribution to the literature. Therefore, we believe the paper to be significantly improved as a result of the comments received from the reviewers.

Editorial comments addressed by the authors 

and 

We thank the editor for pointing out this oversight. Subsequently, we have ensured that PLOS ONE's style requirements are followed throughout the revised submission.

Upon re-submitting your revised manuscript, please upload your study's minimal underlying data set as either Supporting Information files or to a stable, public repository and include the relevant URLs, DOIs, or accession numbers within your revised cover letter. For a list of acceptable repositories, please see http://journals.plos.org/plosone/s/data-availability#loc-recommended-repositories. Any potentially identifying patient information must be fully anonymized.

We thank the editor for pointing out this requirement. Subsequently, we have uploaded our study's minimal underlying data set as supporting information files. Thank you for making the change online on our behalf.

Reviewer comments addressed by the authors 

Reviewer #1: 

The article uses Big Data to study a very important topic of COVID-19 vaccine hesitancy and factors that may enhance vaccine uptake. The authors provide an expansive picture of commonalities and diversities among varied countries, which contribute to the understanding of this vital issue. The article is well-written, maintaining clarity and logical flow in the description of the findings and their potential value.

We thank the reviewer for this kind and generous review of our study.

Nonetheless, there are several elements that need to be improved in order for the article to be appropriate for publication, as follows:

1. The authors' mix' between results and discussion. See section 4.1.2. the results should be presented as found, without the authors trying to provide an explanation to why each element was such found. These explanations (for example see lines 375-388 concerning Belgium or lines 389-396 concerning Netherlands) are not appropriate in the results section.

We thank the reviewer for this comment. Different disciplines have different approaches to the results and discussion section. As well-being economists, we follow the format set out in economic papers in which the results and discussion sections are combined. We prefer to keep it in this format. See similar well-being economics papers using the same structure below (please note that there are many more examples):

Consiglio V, Sologon D. The Myth of Equal Opportunity in Germany? Wage Inequality and the Role of (Non-)academic Family Background for Differences in Capital Endowments and Returns on the Labour Market. Social Indicators Research. 2021. Available from https://doi.org/10.1007/s11205-021-02719-2

Rözer J, Lancee B, Volker B. Keeping Up or Giving Up? Income Inequality and Materialism in Europe and the United States. Social Indicators Research. 2022; 159, 647–666. Available from https://doi.org/10.1007/s11205-021-02760-1

2. The authors should refrain from any judgmental comments (for example lines 395-396 – "It is safe to say that the information campaigns have paid off"). Scientific writing should state possible explanations to any phenomenon that is identified, usually based on previous publications or new theoretical hypotheses, but without any judgment of any kind (neither positive or negative). Furthermore, the authors have not shown any support that this is actually the case and that the information campaigns were the cause for the results identified; more cautious writing is recommended. The same relates to sentences such as line 467 "South Africa's woes are almost too many to count". These types of phrases should be revised so that the sentence presents the finding as is and not referred to in 'literary-type' descriptions.

We thank the reviewer for pointing this out. Subsequently, the entire manuscript has been edited to remove any judgemental comments. Additionally, we have followed a more cautious style of reporting the results.

3. Whenever any assumption (or claimed 'fact') is written – it should be based on previously published data from scientific sources. Sentences such as "Australians have a deep-seated mistrust in their government officials, who seem to implement contradictory regulations across different states" (lines 398-399) have to be supported by such publications.

We thank the reviewer for pointing this out. Subsequently, the entire manuscript has been edited to remove any assumptions reported as fact. Where facts are mentioned, the appropriate references are added. 

4. Lines 505-507 – "Compliance, the act of complying with government-mandated regulations to curb the spread of COVID506 19, is statistically significant and negatively related to the VPAI. Unwillingness to comply likely 507 motivates people to get vaccinated. "is not clear. I would assume that compliance would follow positive attitudes (towards the vaccine or towards the authorized agencies). How do the authors explain the negative correlation?

We thank the reviewer for this comment. We wish to state that we tested this association multiple times and always reached the same outcome of a negative association. We interpret this as when people are unwilling (do not feel like) to comply with orders such as mask-wearing, staying at home etc., then those individuals would be more willing to receive the COVID-19 vaccine, hence a positive attitude (the relationship is negative).

A study conducted by Wright et al. (2022) investigated whether vaccinated individuals were less likely to comply with implemented behavioural measures. The entire premise of the study is that vaccinated individuals believe they are less at risk because of their vaccination status. People think they do not need to comply any longer with, for example, mask-wearing, social distancing etc., if they get vaccinated therefore creating a more positive attitude towards vaccines. 

Wright L, Steptoe A, Mak HW, Fancourt D. Do people reduce compliance with COVID-19 guidelines following vaccination? A longitudinal analysis of matched UK adults. Journal of Epidemiology & Community Health. 2022; 76, 109-115.

5. The authors at times make claims that they do not show any support for based on their data. For example, their claim in lines 534-537 "This implies that, as the tweets related to vaccines increase, the attitude towards vaccines decreases. This may likely be because many of the tweets contain misinformation or conspiracy theories rather than campaigns and information to encourage being vaccinated - and, therefore, decreases positive attitudes towards vaccines.". They should check whether the tweets they investigated actually show this tendency to support such a claim.

We thank the reviewer for this comment. Subsequently, we added to our analysis and performed topic modelling on the extracted vaccine-related tweets. We expanded our explanation on the negative relationship between the number of tweets and the VPAI. Please see lines 169-177 and 590 onwards. We also include an example of the topic modelling in the word version of our response in Table 1 for your perusal. Please note we also added the topic modelling results to explain the trends in VPAI in section 4.1.2. 

Reviewer #2: 

The results reported in the paper are interesting as such, since the topic of public reaction to the COVID-19 pandemic and the related interventions is a timely one. However, to this referee it seems that the descriptive interest of the results dominates over the analytical one, due to various technical issues, as detailed below.

1. Why do authors call the analysis "panel analysis" when there is no panel of individuals followed at more than one time-point? Considering a country as an individual is a complicated assumption... In reality, the data is a sequence of cross-sectional analyses within each country without control of the relative media-related weight or publication frequency of pro- and anti-vax individuals in the data stream and therefore maybe biased as an estimate of the country specific sentiment...

We thank the reviewer for this comment. We wish to point out that the data are a country-level panel dataset. Thus, a macro panel dataset. We are analysing ten countries over time at a daily frequency. Panel datasets can be at a micro, regional or macro level. It only implies that the same unit of analysis (individual/region/country) was measured at different time periods. Please see the following textbooks in this regard:

Baltagi BH. Econometric analysis of panel data. 2005. John Wiley & Sons Ltd: England.

Gujarati DN. Basic Econometrics, (4th ed.). 2003. New York, n. y.: McGraw-Hill.

Additionally, similar to other data on happiness or subjective well-being (survey data), the survey is based on individuals but aggregated to country-level – please see the World Happiness Report.

Helliwell JF, Layard R, Sachs J, De Neve J, eds. World Happiness Report. 2021. New York: Sustainable Development Solutions Network.

With Big Data, each tweet is seen as a source which is then combined to create text corpus per day (thus aggregated to a country level per day) – similar to survey data (aggregated to country level). The number of tweets is vast. We extract 100000 tweets per day for the UK and NZ, the smallest country, 6000 tweets randomly. The sentiment of these tweets is then aggregated to derive a country level sentiment per day. This is a more representative sample than survey data.

2. The analysis methods, common in econometric panel data analyses, have been devised for individual data and, as also commented upon above, have to be motivated for use on aggregated, non-individual, data..

We thank the reviewer for this comment. However, as mentioned in point #1 above, panel analysis at a macro level is well recognised in econometrics. The panel analysis of macro data is included in econometric textbooks. Please see amongst other: 

Breitung J. The Analysis of Macroeconomic Panel Data. 2015. Oxford handbooks. Available from https://www.oxfordhandbooks.com/view/10.1093/oxfordhb/9780199940042.001.0001/oxfordhb-9780199940042-e-15

Gujarati DN. Basic Econometrics, (4th ed.). 2003. New York, n. y.: McGraw-Hill

Specifically, please see page 660 "In panel data the same cross-sectional unit (say a family, a firm, a state or a country) is surveyed over time. In short, panel data have space as well as time dimensions." 

3. One could also suspect that people are more eager to announce an anti-vaccine stance than a neutral or pro-vaccine stance on social media, especially since the anti-vaxers are minorities, which therefore need more cohesion and solidarity...

We thank the reviewer for this comment. Subsequently, we performed topic modelling on the extracted vaccine-related tweets in addressing point #5 from reviewer 1. We did not find what you are alluding to. Please see the example of the topic modelling in table 1 in the word version of our response for your perusal.

4. The paper also seems to conclude that a part of the negative opinions are more due to vaccine availability and related problems than to an "ideological" anti-vaccine stance. A clearer separation of these issues in the analysis, in addition to the VPAI2 index, would help the understanding of vaccine-related sentiment...

We thank the reviewer for this comment. We have made significant changes throughout the discussion of the results, and we trust we have addressed this comment.

5. In section 3.2.2 (Selection of covariates), many covariates are extracted from the same source as the response variable (the Tweet streams). Is there not a risk that this automatically induces correlations?

We thank the reviewer for this comment. The initial text corpus (extracted tweets) indeed are the same. However, we use different keywords to extract tweets from the initial corpus. Please see page 15 for the full description. We summarise below as well. 

In our analyses, we make use of the following variables, which are constructed from tweets: 

1) VPAI: on page 12, we explain that the index is constructed from corpus data extracted from vaccine-related keywords. To construct the index, we use sentiment analysis, in which an algorithm is used to determine the sentiment of the tweets. 

2) Trust in the COVID-19 Vaccine: on page 15, we explain that this variable was constructed using the same corpus of extracted tweets as in 1) (keywords related to vaccines), although, in this instance, we do not use sentiment analysis but the underlying coded trust emotion. Two different algorithms are used based on different principles to analyse sentiment and emotion. The output of the sentiment and emotional analysis differ vastly. Only certain tweets will be coded as containing the trust emotion – whereas almost all tweets are coded as having a sentiment of different degrees.

3) Lagged anger towards the government: on page 15, we explain that the corpus of extracted tweets is based on government-related keywords. Therefore, the corpus tweets differ from the prementioned variables (1) and (2).

Therefore, even if the original corpus of tweets is the same, the coded tweets used to construct the variables are often not the same tweets. Even where tweets may overlap, the algorithms used to derive sentiment (Sentiment140 or Syuzhet, AFINN and Bing for robustness test) or emotion (NRC) differs. Please see the VIF in Table 2 in the word version of our response to show no multicollinearity.

The VIF value (test for multicollinearity) of lagged trust in COVID-19 vaccines is 2.70, and lagged anger towards governments is 4.51. Both are well within the accepted ranges of less than 5.

6. In the equation (2, p13), one wonders what common monthly effect (lambda_m) is expected in different hemispheres, in countries in different phases of epidemic spread...

We thank the reviewer for this comment. The month fixed effect eliminates bias from unobservable variables that change over time but are constant over the countries. Therefore, we control for the heterogeneity in months. 

We extended our explanation in line 314 onwards to clarify this matter. However, we summarise it here as well:

1) seasonal effects (changes in seasons): occur in all countries in the panel in the same months. During March in the Northern hemisphere, the season changes to Spring and in the Southern hemisphere to Autumn. It has been shown that changes in seasons can affect attitudes. Similar in June, the change in season is to Summer and Winter, respectively.

2) evolution of the pandemic: as the pandemic progressed from one month to the next, people's attitudes might have changed. They might have gotten used to the idea of pandemics and vaccines. Or they might show increased pandemic or vaccine fatigue which could also affect their attitudes.

3) holiday seasons: July is a holiday season (Northern hemisphere – Summer holidays and Southern hemisphere- winter holiday).

We find that each month has a different Y-intercept. This implies that certain characteristics of months are the same across countries, but there is heterogeneity between the months. Please see the results in Table 3 in the word version of our response. 

From Table 3, we can see that as the year progressed, compared to February, the Y-intercept per month decreased, and these differences were significant. Therefore, the months are not the same (there is heterogeneity), and these differences are unobserved. The Y-intercept for February is 0.24. Ceteris paribus, we notice that in March, the positive attitude is significantly different to February, which is lower with 0.04. 

7. The analysis in the paper is also to a large extent outdated, since so much , both in terms of restrictions and vaccinations, has happened since August '21...

In case of revision, the authors should incorporate some recent material, either in the analysis or the discussion.

We thank the reviewer for this comment. However, we respectfully disagree. Our analysis is for a specific period for which we attempt to understand what happened to VPAI in specific circumstances. If we understand the past, we are also more likely to understand the implications of future actions. Therefore, our findings based on the past can inform policymakers on the effect their actions might have on attitudes towards vaccines. For example, we find that people unwilling to comply are more positive towards vaccines. This information informs policymakers what to expect from people's attitudes when regulations are implemented.

8. Finally, regarding section 5 (Conclusions), a clearer distinction between conclusions strictly based on the analysis and authors opinions or suggestions for action should be made.

We thank the reviewer for this comment. Subsequently, addressing points #2 and #3 raised by reviewer 1, we believe this point has been addressed. Please note we also added the topic modelling results to explain the trends in VPAI in section 4.1.2. Thereby eliminating any opinions and only reporting results. 

9. Small language errors or typos to be corrected, e.g. "illicit" instead of "elicit" (p2 l36) or "By analysing, the latter..." (p7 l162) instead of "By analysing the latter..."

We thank the reviewer for pointing out these errors. Subsequently, the manuscript was proofread, and any other mistakes were corrected.

Reviewer #3: 

This paper has two main objectives: 1) to conduct a cross-country panel analysis to investigate the trend in positive attitudes towards COVID-19 vaccines over time; and 2) lies with determining those variables which are significantly related to a positive vaccine attitude and can inform policymakers. The article has a good description of how previous studies analysed the emotions in vaccine-related tweets, and following this description, the authors present a considerable justification for why do this study, which is classified as the first study to use Big Data to determine the sentiment and emotions related to COVID-19 vaccines through a vaccine positive attitude index. The time series analysis is a good way to perform a point of view of this situation.

The authors' proposal is to present an analysis based on tweets from 10 countries, also dividing the analysis into countries in the southern hemisphere and countries in the northern hemisphere. This choice is based on data availability, which can be considered a limitation of the study, in view of a non-characterization of how other countries could present differences regarding the information provided.

Minor concerns

The text of the paper needs to be revised as it has some inconsistencies regarding the information.

1. About Figure 1 - it is necessary to modify the dates of the years presented in this figure, considering that the data were collected in 2021, the years change according to the countries. In addition, I also suggest a review of the percentages presented in terms of the number of people fully vaccinated, which differ from Table 1.

We thank the reviewer for pointing out this oversight and the suggestion. Subsequently, figure 1 and table 1 were updated with the fully vaccinated population percentage (31 July 2021).

2. Section 3.2.1: the authors present some illustrations based on word clouds generated by tweets and inform that this was done for all country but in the paper, it's possible to find just 2-word clouds showing information by Great Britain and South Africa. It's important to inform why just these 2 countries were present in this analysis.

We thank the reviewer for this comment. On page 12, we state that word clouds were constructed for all countries. However, we use Great Britain and later South Africa purely as examples of the word clouds. The manuscript already has twelve figures and three tables, so adding an additional eight word cloud figures becomes impractical. 

3. Section 3.2.2: Scale 5c. In the second paragraph, the word complacency is not present.

We thank the reviewer for pointing out this oversight. This has been corrected.

4. The authors do not present the limitations of the study, an important point for future studies on the subject. In view of some methodological characteristics, my suggestion is to highlight some points such as: the translation of tweets in a non-English language; choice of countries based on data convenience and availability.

We thank the reviewer for this suggestion. Subsequently, the study's limitations have been added at the end of the conclusion section.

5. There is no specific topic in the study for discussing the data, however, in the description of the results, the authors present some references that can be considered part of this discussion of the data. As this is an analytical study, it would be important to have a more in-depth discussion, comparing the results to other previous studies.

We thank the reviewer for this comment. We wish to point out that, as you state in point #6, our results are discussed based on the theoretical background provided in section 3.2.2 and the literature review provided in section 2. Furthermore, we have made changes throughout the discussion of our results referring to previous studies.

6. The manuscript has good methodological quality, is free of bias, needs some adjusts but the results are discussed based on the theoretical background properly of the manuscript theme. So, the conclusions answered the aims of the study focused on the references and results. The limitations of the study were not presented, and I suggest that this could be done.

---

## [Decision Letter · Decision Letter 1]

22 Feb 2022

Positive attitudes towards COVID-19 vaccines: A cross-country analysis.

PONE-D-21-34544R1

Dear Dr. Rossouw,

We’re pleased to inform you that your manuscript has been judged scientifically suitable for publication and will be formally accepted for publication once it meets all outstanding technical requirements.

Kind regards,

Prof. Anat Gesser-Edelsburg, Ph.D.

Academic Editor

PLOS ONE

Additional Editor Comments (optional):

Reviewers' comments:

Reviewer's Responses to Questions

**Comments to the Author**

1. If the authors have adequately addressed your comments raised in a previous round of review and you feel that this manuscript is now acceptable for publication, you may indicate that here to bypass the “Comments to the Author” section, enter your conflict of interest statement in the “Confidential to Editor” section, and submit your "Accept" recommendation.

Reviewer #1: All comments have been addressed

Reviewer #2: (No Response)

Reviewer #3: All comments have been addressed

2. Is the manuscript technically sound, and do the data support the conclusions?

Reviewer #1: Yes

Reviewer #2: (No Response)

Reviewer #3: Yes

3. Has the statistical analysis been performed appropriately and rigorously? 

Reviewer #1: Yes

Reviewer #2: (No Response)

Reviewer #3: Yes

4. Have the authors made all data underlying the findings in their manuscript fully available?

Reviewer #1: Yes

Reviewer #2: (No Response)

Reviewer #3: Yes

5. Is the manuscript presented in an intelligible fashion and written in standard English?

Reviewer #1: Yes

Reviewer #2: Yes

Reviewer #3: Yes

6. Review Comments to the Author

Reviewer #1: The article adds important input to readers; the authors appropriately modified the manuscript and addressed all comments

Reviewer #2: The authors replied to all the reviewer's comments and edited the manuscript in various areas. The paper may be published as is, though some weaknesses remain, for example the debate on the representativeness of tweets to reflect the feelings of the general population, including those who do not use tweets.

Reviewer #3: (No Response)

7. PLOS authors have the option to publish the peer review history of their article (what does this mean?). If published, this will include your full peer review and any attached files.

Reviewer #1: No

Reviewer #2: No

Reviewer #3: No

---

## [Editor Report · Acceptance letter]

2 Mar 2022

PONE-D-21-34544R1 

Positive attitudes towards COVID-19 vaccines: A cross-country analysis. 

Dear Dr. Rossouw:

I'm pleased to inform you that your manuscript has been deemed suitable for publication in PLOS ONE. Congratulations! Your manuscript is now with our production department. 

Kind regards, 

on behalf of

Prof. Anat Gesser-Edelsburg 

Academic Editor

PLOS ONE